# Plant Extracellular Vesicles: Current Landscape and Future Directions

**DOI:** 10.3390/plants12244141

**Published:** 2023-12-12

**Authors:** Alfredo Ambrosone, Ani Barbulova, Elisa Cappetta, Fabrizio Cillo, Monica De Palma, Michelina Ruocco, Gabriella Pocsfalvi

**Affiliations:** 1Department of Pharmacy, University of Salerno, 84084 Fisciano, Italy; aambrosone@unisa.it (A.A.); ecappetta@unisa.it (E.C.); 2Institute of Biosciences and BioResources (IBBR), Research Division (R.D.) Naples, National Research Council of Italy (CNR), 80131 Naples, Italy; gabriella.pocsfalvi@ibbr.cnr.it; 3Institute for Sustainable Plant Protection, Research Division (R.D.) Bari, National Research Council of Italy (CNR), 70126 Bari, Italy; fabrizio.cillo@ipsp.cnr.it; 4Institute of Biosciences and BioResources (IBBR), Research Division (R.D.) Portici, National Research Council of Italy (CNR), 80055 Portici, Italy; monica.depalma@ibbr.cnr.it; 5Institute for Sustainable Plant Protection, Research Division (R.D.) Portici, National Research Council of Italy (CNR), 80055 Portici, Italy; michelina.ruocco@ipsp.cnr.it

**Keywords:** extracellular vesicles, apoplastic vesicles, root exudate, plant in vitro tissue and cell culture

## Abstract

Plant cells secrete membrane-enclosed micrometer- and nanometer-sized vesicles that, similarly to the extracellular vesicles (EVs) released by mammalian or bacterial cells, carry a complex molecular cargo of proteins, nucleic acids, lipids, and primary and secondary metabolites. While it is technically complicated to isolate EVs from whole plants or their tissues, in vitro plant cell cultures provide excellent model systems for their study. Plant EVs have been isolated from the conditioned culture media of plant cell, pollen, hairy root, and protoplast cultures, and recent studies have gathered important structural and biological data that provide a framework to decipher their physiological roles and unveil previously unacknowledged links to their diverse biological functions. The primary function of plant EVs seems to be in the secretion that underlies cell growth and morphogenesis, cell wall composition, and cell–cell communication processes. Besides their physiological functions, plant EVs may participate in defence mechanisms against different plant pathogens, including fungi, viruses, and bacteria. Whereas edible and medicinal-plant-derived nanovesicles isolated from homogenised plant materials ex vivo are widely studied and exploited, today, plant EV research is still in its infancy. This review, for the first time, highlights the different in vitro sources that have been used to isolate plant EVs, together with the structural and biological studies that investigate the molecular cargo, and pinpoints the possible role of plant EVs as mediators in plant–pathogen interactions, which may contribute to opening up new scenarios for agricultural applications, biotechnology, and innovative strategies for plant disease management.

## 1. Introduction

Extracellular vesicles (EVs) are biomembrane-enclosed heterogeneous structures that are ubiquitously secreted by living cells. By carrying functional molecules, such as proteins, oligonucleotides, and lipids, between close and distant cells, EVs participate in many physiological and pathological processes. Due to increased interest in their therapeutic and diagnostic potential, research on human-cell- and cell-culture-derived EVs is very active [1]. Several studies show evidence that plant cells also secrete EVs similar to mammalian cells in morphology [2]. Membranous vesicular structures called “paramural bodies” were observed between plant cell walls and cell membranes already in the 60s [3,4]. In an early electron microscopy study, Halperin and Jansen, amongst the first, reported the presence of small vesicles, similar to exosomes both in location and in morphology, released by the multivesicular bodies (MVBs) during the embryogenesis of wide carrot cells in a suspension culture [4]. Later, it was further proven that plant cells can secrete EVs into the extracellular space via the fusion of MVBs with the plasma membrane [5] giving rise to studies that tackled their roles in cell wall remodelling, unconventional protein secretion, RNA transport, defence against pathogens, and plant–microbe symbiosis [6].

In parallel with the research on mammalian EVs, several articles have reported on the anti-inflammatory, antitumoral, and antioxidant effects of different plant-derived nanovesicles (PDNVs) in vitro and in vivo [7,8,9,10,11,12,13], and ongoing research tries to find suitable sources for their reproducible and large-scale production. PDNVs can be isolated with high yield directly from the juice of the plant; after homogenisation, from plant organs like fruits, roots, and leaves; or even from whole plants (Figure 1). Due to their green origin, PDNVs are currently under extensive investigation for potential applications in the pharmaceutical, nutraceutical, and cosmeceutical industries, both as pristine nanomaterials and as delivery vectors. The quality, safety, and reproducibility of PDNV isolates, however, are not always satisfactory. For example, it has been shown that tomato NVs are frequently, if not always, associated with various viruses [13], and NVs isolated from strawberries contain allergens [14]. In addition, PDNV isolates are very complex mixtures. Besides EVs, they contain intracellular vesicles, obtained as a result of cell rupture during homogenisation and various vesicles, that are formed during different steps of the isolation process [15]. With the current analytical methods, it is not possible to isolate the EV populations from a PDNV sample.

True EVs which are present in the extracellular milieu have been isolated from the following plant sources (Figure 1): (i) apoplastic fluid [6,8,11,16,17,18,19,20,21,22,23,24,25,26,27,28,29,30,31,32,33,34,35,36,37,38], (ii) exudates [21], and (iii) conditioned culture media (CCM) [4,22,23,38,39,40,41,42,43]. Meanwhile, several recent reviews focus on the isolation and features of edible-plant-derived nanovesicles, outlining their promising biological properties [8,9,11]. In this review, we turned our attention towards plant EVs isolated from in vitro cell suspensions and pollen and hairy root cultures, as well summarising findings related to their biocargo composition and role in plant–pathogen interactions. We would like to emphasise that if any structural and functional differences exist between EVs and apoplastic vesicles (AVs, including paramural and extramural vesicles), they have not yet been clearly defined so far. However, in the following paragraphs, we have clearly indicated whether the studies were conducted with EVs or AVs.

## 2. In Vitro Plant Cultures as a Source of EVs

In vitro plant cell and tissue cultures (Figure 1) certainly represent a valid green alternative to mammalian CCM for EV production. Plant cell suspension cultures (CSCs), as well as hairy root cultures (HRCs), are largely used as bio-factories of commercially interesting active ingredients for the pharmaceutical and cosmetic industries. They offer standardisable, scalable, contaminant-free, and bio-sustainable systems that allow the production of the desired compounds to be extended to an industrial scale [44]. Moreover, due to the high plasticity of the plant cells, which distinguishes them from all living organisms, plant cultures are an extremely versatile system. Plasticity is the ability of the plant cells to change metabolism and switch on different biosynthetic and development pathways, adapting their growth to the surrounding conditions. This makes it possible to change the culture conditions, elicit the cells in a controlled manner using genetic or biochemical tools, and “guide” them to the production of EVs involved in physiological and defence mechanisms. While plant molecular farming technology is used for the production of various molecules, such as proteins and secondary metabolites [45], today, it is only limitedly exploited for EV harvesting. Here, we will summarise the current findings and outline some possible next steps towards the establishment of plant in vitro cultures for EV studies or production.

### 2.1. Plant Cell Suspension Culture-Derived EVs

Despite the early observations of the existence of plant EVs being made in a carrot cell suspension [4], today, there are only a few species, tobacco (*Nicotiana tabacum* L.) [22,23], Korean ginseng (*Panax ginseng* C.A. Meyer) [40], and two flowering plants, blue carpet (*Crate rostigma plantagineum* Hochst.) [23] and kalimeris (*Aster yomena*) [46], from which EVs have been isolated. Woith and co-workers, for the first time, isolated EVs from tobacco and blue carpet [23]. The authors cultured the cells in 2 L flasks for two weeks and used CCM as the EV source material. For the isolation, they used combined filtration and differential centrifugation at a maximum velocity of 50,000× *g* (note: the gold-standard differential ultracentrifugation (dUC) method generally used for the isolation of mammalian EVs employs 100,000× *g*) or tangential flow filtration (TFF) methods. Transmission electron microscopy (TEM) of the TFF-isolated tobacco EVs showed cup-shaped vesicles and other co-purified structures. Using this technique, no morphological differences could be observed between the EVs of the two species. SDS-PAGE gel and LC-MS/MS in-solution-based shotgun proteomic analysis of the EV isolate (this time, only for *C. plantagineum*) identified 35 proteins common in the two replicates of the EV isolate, and revealed the presence of enzymes involved in cell wall remodelling (galactosidase, glucosidase, pectin esterase, arabinofuranosidase, etc.), confirming the hypothesis that they could be an active part of the secretion mechanism of EVs through the cell wall. Additionally, some transmembrane proteins such as transmembrane 9 superfamily member 11, adaptor protein complex subunits, and membrane steroid-binding protein 2, as well as proteins related to ubiquitination, were identified, indicating the possible plasma membrane origin of the EVs. Moreover, the presence of endochitinase EP3-like protein, important for defence against pathogenic fungi, was also observed. The small amount of the isolated EVs apparently did not allow the phospholipid and secondary metabolite analyses that the authors performed on the PDNV extracts.

Kocholata et al. compared the efficiency of differential ultracentrifugation (dUC) and polyethylene glycol (PEG) precipitation for the isolation of EVs from *Nicotiana tabacum* BY-2, a highly used model system in the field of plant tissue culture [22]. The cells were grown for 7 days and 15 mL of CCM was used for the isolation. The PEG precipitation method was quicker than dUC but it resulted in the co-isolation of the proteins. The EVs tended to aggregate when isolated, especially in the dUC process due to their negative zeta potential, in contrast with PEG precipitation, where vesicle aggregation was not observed. To prevent aggregation, the effect of trehalose was studied at different stages of the dUC isolation, but it was not possible to eliminate this problem with complete efficiency. Additionally, the authors evaluated the biochemical and biophysical properties of the EVs and confirmed their ability to enter mammalian rat mesenchymal stem cells and BY-2 tobacco cells. The authors also performed isolation of the PDNVs from callus culture homogenate. Interestingly, they found similarities in the characteristics of the CCM EVs and callus PDNVs.

Cho et al. isolated EVs from the supernatant of *Panax ginseng* cells cultured in a 2 L bioreactor for 2 weeks using the dUC method [40]. Tuneable resistive pulse sensing (TRPS) analysis showed small EVs with a mean diameter of 72 ± 25.95 nm, and the concentration was 6.94 × 10^10^ particles per mL of CCM. Based on the results, the authors concluded that the ginseng cells grown in suspension culture released vesicles that possessed anti-senescence and anti-pigmentation activities against UVB-induced senescent melanocytes, underling the importance of suspension cultures as a sustainable source of EV production and purification. Importantly, the ginseng EVs had a distinct lipid profile compared to ginseng root-derived nanovesicles and were shown to be enriched in diacylglycerols, phospholipids (phosphatidylcholine, phosphatidylethanolamine, lysophosphatidylcholine), and sphingomyelin, revealing their unique vesicular properties.

Kim et al. sourced EVs from the supernatant of tissue culture of a two-week-old *Aster yomena* using a combination of differential centrifugation, tangential flow filtration, and cushion ultracentrifugation [46]. This study, besides physiochemical, morphological, and biocargo characterisation, aimed at investigating the in vivo effect of EVs of *A. yomena* on human health. EVs injected into mouse models of asthma were found to reduce the levels of major factors for developing asthma. Importantly, this paper highlights several possible advantages of in vitro-produced plant EVs in the development of novel therapeutics for the treatment of different inflammatory diseases.

### 2.2. Callus-Culture-Derived Vesicles

Callus culture is the technique of growing plant tissues or cells in vitro in an artificial medium, usually containing relatively high auxin concentrations or a combination of auxin and cytokinin [47]. Attempts have also been made to isolate vesicles from plant callus cultures of two species, tobacco [22] and *Arabidopsis thaliana* [48]. In these studies, the harvested calli were homogenised, and vesicles were isolated from the homogenisation buffer. A comparative analysis shows fewer callus-derived vesicles (1.8 × 10^10^) compared to apoplastic-fluid-derived AVs (2.9 × 10^10^ particles g^−1^ fresh weight) in Arabidopsis [48]. Based on scanning electron microscopy (SEM) and nanoparticle tracking analysis (NTA), the callus-derived vesicles showed a rather broad size distribution, and the expression levels of endosomal sorting complex (ESCRT) related TET8 and PEN genes were found to be significantly lower too. According to our view, mechanical homogenisation can lead to the release of secretory vesicles, transport vesicles, AVs, and EVs into the homogenisation media, which complicates the understanding of the functional role of single vesicle types. It would be very interesting to deepen our understanding of the secretion of EVs in transdifferentiation, callus formation, and somatic embryogenesis, but adequate tools need to be developed for the isolation of EVs from plant tissues.

### 2.3. Protoplast Culture

Plant cultures are extremely suitable systems to produce not only bioactive compounds with industrial applications but also provide tools to study different physiological processes at the single-cell level. While CSCs are complex systems, protoplast cultures harbour a more homogeneous cell population, lacking both cell walls and physical connections between neighbouring cells [44]. Interestingly, highly dynamic exo- and endocytic events were observed in a *N. tobacum* BY-2 cell-derived protoplast [39]. The authors analysed the kinetics and size of single exo- and endocytotic events in BY-2 protoplasts in real time by revealing the proportional changes in the electrical membrane capacitance. These results lay the foundation for future investigations. Furthermore, the authors concluded that the kinetics of fusion and fission appear to be similar in both animal and plant cells, suggesting highly conserved mechanisms among eukaryotes.

### 2.4. Hairy-Root-Culture-Derived EVs

Only one study has reported so far the successful isolation of EVs from the conditioned medium of HRC [42]. EVs were isolated from the HRC of *Salvia dominica* using dUC. The dynamic light scattering (DLS) method, combined using NTA and TEM, revealed that the hairy roots of *S. dominica* secrete lipid-membrane-enclosed, round-shaped EVs ranging in size between 100 and 200 nm. TET-7, a tetraspanin with high homology to the plant exosomal marker TET-8, was also identified in the HR-derived EVs. The authors also examined the uptake of the isolated EVs by human cells, human keratinocytes (HaCaT cells), and pancreatic carcinoma cells (MIA PaCa-2), and demonstrated that the HR-derived EVs of *S. dominica* mostly entered into and accumulated in the cytoplasm, preferentially in cancer cells, showing strong and selective antiproliferative and pro-apoptotic activity.

### 2.5. Pollen-Culture-Derived EVs

Pollen is a microspore formed by seed plants and used as a source to produce haploid plants. Prado and co-workers showed the release of EVs, termed pollensomes, during pollen germination and pollen tube growth [38,43]. The EVs were isolated using consecutive filtration and ultracentrifugation from the germination medium of in vitro-growing pollen tubes of olive (*Olea europaea*), and further separated into fractions using sucrose gradient ultracentrifugation. SEM of the isolates revealed a heterogeneous population of small (28 to 60 nm in diameter) and round-shaped EVs [38]. Pollensomes have a high density (1.24 and 1.29 g mL^–1^) compared to mammalian EVs. Fourier-transform infrared (FTIR) analysis showed that the pollensomes, besides lipids and proteins, were enriched in polysaccharides and pectin. MS-based proteomics highlighted the presence of many proteins, several of them participating in metabolism and signalling, cell wall expansion, and membrane transport biological functions, suggesting that pollen EVs may have a role in pollen tube growth, pollen–stigma interactions, and in general in fertilisation. Moreover, olive pollensomes were shown to contain allergens, such as Ole e 1, Ole 5 e 11, and Ole e 12, implying their potential involvement in allergic reactions [43]. More recently, EVs from germinating kiwi (*Actinidia chinensis)* pollen culture were also isolated and, for the first time, it was shown that the vesicles contain a double-layer biomembrane. An in-depth proteomic characterisation revealed the presence of typical EV markers such as tetraspanins (tetraspanin-8 like and tetraspanin-15 like), ESCRT-related proteins, HSPs, and Bro-1domain containing protein (the plant homologue of apoptosis-linked gene-2 interacting protein X, ALIX) [41].

Taken together, these studies underline the successful isolation of plant EVs from different in vitro plant cultures using CCM as the starting material. The preparations obtained so far contained small round-shaped membrane-enclosed vesicles, easily forming aggregates, and containing protein and lipid cargo. Given these findings are state-of-the-art, it is difficult to assess the reproducibility of the isolation or compare the yield and biocargo composition of the systems studied. Moreover, no RNA and metabolite cargo have yet been systematically studied in plant in vitro-culture-derived EVs. An accurate description of the characteristics of the source mother cells, as well as culturing parameters for the production of EVs, has not been implemented in all publications, and studies on how the cell growth conditions influence the EVs’ characteristics and the role of EVs in plant physiology are still to be undertaken.

## 3. Insights into Plant Extracellular Vesicles Biogenesis and Release

The first discoveries of membrane-associated vesicles date back to the late 1950s.

“Border bodies” was the term to indicate vesicular structures associated with the plasmalemma in the fungus *Polyporus versicolor* [49]. Since then, different vesicular structures associated with plasmalemma, generally termed “paramural bodies” or “lomasomes” have been found in numerous species, including algae and higher plants [50,51]. It is worth mentioning that for a long time, paramural bodies were considered potential fixation artifacts. However, the advent of advanced imaging techniques, including transmission electron microscopy (TEM), SEM, cryo-electron microscopy, and atomic force microscopy (AFM), has gradually contributed to describing the morphology of these vesicular structures and elucidating, at least in part, the mechanisms of EV biogenesis and release.

To date, several studies have revealed the existence of different routes involved in EV or AV biogenesis and their cellular release in plants; however, not all of them have been fully elucidated [6,16,20,26,30]. Among the known biogenesis pathways described in the literature, multivesicular body (MVB)–plasma membrane (PM) fusion is the most studied and well characterised (Figure 2A).

MVBs, or multivesicular endosomes, are small spherical organelles containing internal vesicles. They were first described in rat oocytes using electron microscopy [52,53]. MVBs primarily serve as intermediate protein sorting and processing compartments between the trans-Golgi network (TGN) and lytic vacuoles (lysosomes) or storage vacuoles [52].

In contrast to mammals and yeasts, the transformation of MVBs is thought to occur via the maturation of clathrin-coated tubular networks in the Golgi stack matrix, resembling the role of early endosomes (EE) in mammalian cells. Some studies suggest that extracellular vesicles (EVs) may originate from the TGN/MVB [54,55]. The first evidence, from as early as the 1960s, reported that MVBs can fuse with the plasma membrane (PM), releasing their intraluminal vesicles (ILVs) into the extracellular space in chemically fixed carrot cells [4]. More recently, this phenomenon has also been described in other plant cells, confirming the key role of MVBs in biogenesis [5,17]. EV release via the MVB is well described in mammalian cells as well. It is controlled by a multi-protein ESCRT machinery, which controls MVB and ILV formation and the sorting of ubiquitinated membrane proteins for the endosomes [17,56]. In more detail, ESCRT consists of four different protein complexes, including ESCRT-0, -I, -II, and -III, and accessory proteins (like the AAA ATPase Vps4 complex) that bind and sequester ubiquitinated proteins, sorting them into the intraluminal vesicles of multivesicular bodies [57]. Each component of the ESCRT machinery is conserved in plants, with the only exception being ESCRT-0, an early-acting protein complex involved in the initial targeting of the ubiquitinated cargo and recruitment of the downstream protein complexes ESCRT I, II, and III [57,58,59]. To fill this gap, over the last decade, several ubiquitin-binding proteins have been identified and suggested as plant-specific substitutes for the ESCRT-0 subunit. Among the putative candidates, the FYVE domain protein required for endosomal sorting 1 (FREE1) has been suggested to replace the role of the ESCRT-0 complex by interacting and colocalising with the ESCRT-I complex via its VPS23 [58,60,61]. Another candidate for ESCRT-0 is the orthologue of mammalian TOM-1, which probably evolved via duplication events, as suggested by the nine TOM1-like (TOL) ubiquitin receptor proteins identified in Arabidopsis [59]. Owing to their ability to localise in early endosomal structures and recognise and sort ubiquitinated cargo closer to the plasma membrane, it is believed that they may replace the ESCRT-0 machinery in plants [62,63]. Even though these efforts in recent years have confirmed the role of MVBs in plant EV biogenesis and revealed potential plant ESCRT genes, a mechanistic link between MVBs and EVs is still lacking. Recently, thanks to the emergence of the first plant EV biomarkers like TET8 and TET9 tetraspanins, and the syntaxin PENETRATION1 protein (PEN1), this link could be established [24]. However, although the co-localisation of AtTET8 has been shown with the known MVB marker ARA6 [64,65], the other biomarker PEN1 did not co-localise with ARA6, suggesting the theory that the PEN1- and TET8-positive EVs have distinct biogenesis pathways [64,66]. Furthermore, there is recent evidence that these latter also fractionate differently and differ in their vesicular cargo [67]. Noteworthy, protein biomarkers of EVs may not only link EVs to secretion pathway(s) and interaction with the plasma membrane but also enable the quantification, purification, and in situ localisation of EVs, and provide a promising starting point for studying their molecular cargo, which in turn will provide information about the biogenesis pathway(s) and their biological functions.

In addition to the MVB pathway described above, there is also evidence for alternative EV biogenesis in plants, including exocyst-positive organelle (EXPO)-mediated secretion, autophagosome-mediated secretion, and vacuole–PM fusion (Figure 2A).

EXPOs are novel spherical double-membrane organelles like autophagosomes [68], discovered by expressing an Arabidopsis homologue of the exocyst protein Exo70 (Exo70E2) in Arabidopsis and tobacco cells [68,69]. EXPOs can fuse to the plasma membrane and release single-membrane cargo-containing vesicles, harbouring the protein marker Exo70E2 [69]. However, despite their similar morphology, EXPOs and autophagosomes do not co-locate together, so these two organelles are considered distinct entities, as suggested by Ding et al. [69]. Although little is known about the biological significance of EXPO-mediated EV secretion in plants, their characteristics have recently been revised [6], underlying their possible role in the release of exosomes containing leaderless proteins involved in growth regulation and remodelling of the plant cell wall [70].

In addition to MVBs and EXPOs, vacuoles and autophagosomes may also contribute to the biogenesis of certain EV classes in plants. In this regard, a recent study by Cui and colleagues [71] revealed that intraluminal vesicles may be present in small vacuoles, originating from their fusion with MVBs, thus suggesting that some EVs may originate from the fusion of vacuoles with the membrane (Figure 2A).

The other source of EV biogenesis is the autophagosome, well studied in mammals and yeast, which can fuse with MVBs to form amphisomes, which in turn are integrated into the PM, secreting their contents [72,73] (Figure 2A). Zhao et al. suggested that a similar process could occur also in plants since the authors supposed that autophagosomes may fuse with MVBs to form “amphisomes” [74]. However, their role in plant EV biogenesis and secretion has not been characterised thoroughly. Besides the above-described EV secretion mechanisms, and similarly to mammalian cells, membrane blebbing could also occur in plants, but this mechanism has not been yet elucidated.

Once EVs have been transported across the plasma membrane, part of them migrates through the cell wall [65] (Figure 2B). Physically, an EV particle is too large to fit through the dense cell wall network made of lignin, pectin, and hemicellulose fibrils, which forms a natural barrier for EVs in plant cells. Due to the cell wall stiffness, the idea that plant EVs could be released/taken up and thus be involved in plant biological processes has long been challenged. Although some progress has been made in recent years, to our best knowledge, only a few studies have demonstrated the possibility of EV secretion crossing the cell wall [21]. Recent studies have revealed the presence of different cell wall hydrolases such as 1,3-β-glucosidases, pectinesterases, polygalacturonases, β-galactosidases, and β-xylosidase/α-L-arabinofuranosidase 2-like in EV-derived proteins, such as [23,31,74], paving a new opportunity for the temporary destabilisation of the cell wall structure, which might facilitate the transition of EVs through the cell wall. It remains to be clarified whether cell wall hydrolases are membrane-associated and whether they effectively retain their enzymatic activity and facilitate cell wall passage.

It is noteworthy that since EVs are lipidic structures containing various subsets of lipids, it could be possible that some of them impart special fluidity properties that allow EVs to compress while moving through cell wall pores. Indeed, even though plant EVs typically appear as spherical structures, tubular and distorted shapes are also conceivable, as previously observed in bacteria, which facilitates cell wall passage [75].

It is also reasonable to assume that EVs can overcome the cell wall thickness using passive diffusion under all conditions that induce subtle changes in the cell wall mechanical properties or compromise its stability, such as development (e.g., morphogenesis at the cell and tissue level), abiotic stresses, and infection [76]. Interestingly, Movahed et al. clearly documented EV passage through the cell wall in *Nicotiana benthamiana* leaves infected with turnip mosaic virus (TuMV), suggesting that viral proteins and RNA spread in the extracellular space as replication complexes within EVs [18].

## 4. Plant EV Biocargo

Plant vesicles (EVs, AVs, and PDNVs) contain complex and dynamic biocargo, comprising proteins, lipids, oligonucleotides, as well as primary and secondary metabolites. When delivered to target cells, the biocargo can trigger different cellular responses, often referred to as cell-to-cell communication. Plant EVs have been shown to have a role in (i) immune response to invading pathogens [18], (ii) plant–microbe interactions [6,30], and (iii) cell wall organisation [35]. Several proteomic, lipidomic, metabolomic, and RNA analyses have been performed on EVs and AVs isolated from different plant species since the cargo composition is indicative of the putative function of the EVs. The paragraphs below and Table 1 summarise these efforts.

### 4.1. Proteins in Plant EVs

Most of the biocargo studies have been focused on the proteomic profiling of plant EVs. Proteomic analysis highlighted the enrichment of AVs isolated from the apoplastic fluids of intact leaves (*Arabidopsis thaliana*) or seedlings (*Helianthus annuus*) in proteins associated with extracellular function, such as glycosylphosphatidylinositol-anchored proteins (GPI-APs), membrane-trafficking proteins (e.g., SNAREs soluble N-ethylmaleimide-sensitive factor attachment protein receptors), the syntaxin PENETRATION-1 (PEN1), and the cytosolic proteins Pattelin-1 and -2 (PATL-1 and PATL-2) [25,37,77]. A comprehensive characterisation of proteins from various types of EVs, as discussed in Pinedo et al. [80], revealed their potential utility as markers for distinguishing different classes of EVs and for excluding cellular debris contaminants. In addition, survey of the protein content in plant AVs revealed that most proteins are involved in biotic and abiotic stress responses. In the AVs of *Arabidopsis* leaves, an abundance of defence/stress proteins (e.g., heat shock protein 70, HSP70), RPM1-Interacting Protein 4 (RIN4), and other proteins involved in reactive oxygen species (ROS) signalling and oxidative stress responses (Phospholipase Dα, Annexin1, Ascorbate peroxidase1, and Gluthatione S-transferase) were detected. Similarly, Prado et al. confirmed the presence of defence/stress proteins, including HSP70 and cell-wall-related proteins (e.g., pectinmethylesterase) in EVs collected from the germination medium of olive pollen grains [38]. All these results allow us to strengthen the idea that plant EVs play a role in plant immunity, as well as in remodelling of the cell wall [35,81].

Recent studies have focused on setting up systems for the isolation of EVs from tomato root exudate grown hydroponically [21] and HRC of *S. dominica* [42]. The protein cargo characterisation of tomato-root-derived EVs confirmed the presence of typical protein families associated with both plant and animal EVs (e.g., 14-3-3 protein family, actin, calmodulin, annexins, aquaporins), as well as revealing several proteins involved in plant defence mechanisms such as the hypersensitive-induced response protein 1, a germin-like protein (GLP), subfamily 1 member 19, the monocopper oxidase-like protein SKU5, and endochitinases. Interestingly, the tomato-root-released EVs showed bioactivity against fungal pathogens. HRC-derived EVs contain numerous proteins homologous to proteins present in mammalian EVs, such as actin, tubulin, and kinesin proteins, chaperone proteins like HSP70s, HSP 80, and HSP90, glycolytic enzymes including enolases and glyceraldehyde-3-phosphate dehydrogenases, together with tetraspanin-7 (TET7), which exhibits high homology to *A. thaliana* TET8. *S. dominica* HR-derived EVs have been shown to have a selective and strong pro-apoptotic activity in pancreatic and mammary cancer cells, which opens up new avenues for their exploitation against human cancer.

### 4.2. Lipids in Plant EVs

Lipids in vesicles provide structural stability, protect cargo, and facilitate membrane fusion with target cells. Some lipids act as signalling molecules, influencing further biological processes of recipient cells. Despite this important role, the knowledge on plant EV lipid composition is still limited and has only been reported for AVs. Regente et al. [27] performed the first lipid content analysis of AVs purified from the apoplastic washing fluid of imbibed sunflower seeds and identified phosphatidic acid (PA), phosphatidylinositol (PI) as a major lipid species, and phosphatidylethanolamine (PE) and phosphatidylcholine (PC) as minor lipid components. Recently, a more comprehensive lipid map of the AVs of Arabidopsis rosettes was also provided. This analysis showed that the lipidomes of seed- and leaf-derived AVs were considerably different, with a significant enrichment in the percentage of sphingolipids (~46%) in seed AVs, which was extremely low in the leaf AVs (~0.5%). Among the four classes of sphingolipids, glycosylinositophosphoceramides (GIPCs) accounted for up to 99%. Interestingly, the administration of exogenous GIPCs onto rosette leaves enhanced the AV production by ~50%, suggesting that GIPC levels influence AV biogenesis in Arabidopsis. Further studies on the lipid composition of plant AVs could greatly improve our knowledge on their biogenesis and roles in intercellular communication.

### 4.3. Small RNA in Plant EVs

Small RNA (sRNA) is a major player in bidirectional cross-kingdom RNA interference (RNAi) [82]. In this phenomenon, complex RNAi machinery facilitates the delivery between organisms of the sRNA produced by both the host and the pathogen to modulate the outcome of infection [16]. In-depth characterisation of AVs’ biocargo using advanced omics approaches has shown that plants can release vesicles into the apoplastic space containing RNA-binding proteins and sRNA, some of them involved in response to pathogens infection [37,64,82]. Interestingly, sRNA profiling of AVs isolated from uninfected Arabidopsis leaves highlighted the abundance of RNA molecules named tiny RNA, which are shorter (10–17 nucleotides in length) than canonical microRNAs (miRNAs) and siRNAs [78]. However, the function of these molecules and whether tiny RNA is delivered into pathogen cells via AVs is not yet clear.

Intriguingly, sRNA profiling of fungal cells revealed that plant AVs enter these cells and deliver functional miRNAs able to knock down crucial fungal genes responsible for pathogenicity [64,77]. Similarly, AVs transported a pool of secondary small interfering RNA (siRNA) derived from pentatricopeptide repeat protein (PPR) gene clusters into an oomycete pathogen *Phytophthora capsici*. Notably, it has been experimentally demonstrated that this PPR-siRNA possesses the capacity to effectively silence target genes within the pathogen [28].

These findings further strengthen the involvement of plant AVs in plant defence and inter-kingdom communication [66] and have raised new questions regarding the mechanism of localisation, stability, selection, and loading in AVs, as well as translocation of AVs’ sRNAs content into target organisms. The current studies suggest that trans-species movement of plant AVs’ sRNA probably requires a selective sorting mechanism [67,81]. Several RNA helicases (RH11 and RH37), annexins (ANN1 and ANN2), and ARGONAUTE1 (AGO1) have been identified in AVs during *B. cinerea* infection in Arabidopsis [67]. Given that apoplastic AGO1, RH11, and RH37 resulted specifically in association with EV-enriched sRNA and that the *ago1, rh11rh37* mutants showed a remarkable reduction in sRNA secretion in AVs, the authors reasonably propose that RNA-binding proteins may function in sRNA loading and/or stabilisation [67].

Further omics data to distinguish the diverse RNA species delivered by EVs could greatly improve our knowledge on these specific biomolecules. Interestingly, Karimi et al. demonstrated that apoplastic fluid contains diverse RNA species, such as sRNA and long non-coding RNAs (lncRNA), including circular RNA (circRNA) located outside the vesicular structures and associated with RNA-binding proteins [33]. However, the function of these molecules in cell-to-cell communication or as a component of the immune system is still far from being defined.

### 4.4. Metabolites in Plant EVs

In addition to proteins, lipids, and RNA, the secondary metabolites of plant vesicles could give further information on their roles in plant physiology. The packaging of secondary metabolites in EVs might be a central aspect of plants’ defence against pathogenic microorganisms too. While metabolomics studies on PDNVs have reported the presence of primary and secondary metabolites of high biological interest [16,19,32], current knowledge on the metabolites transported by plant EVs is scant.

Woith et al. [23] investigated the metabolite profiles of plant EVs isolated from different sources (i.e., the apoplastic fluid, cell culture media, homogenised plant materials) of various plant species using high-performance thin-layer chromatography. The analysis revealed that especially lipophilic molecules were vesicle-associated while alkaloids, phenols, and phenylpropanoids were not detected. On the basis of these results, the authors also suggest that the loading of metabolites in plant EVs could be a passive mechanism, rather than an active packaging process. Nonetheless, the identification of ATP-binding cassette (ABC) transporters in plants hints at the possibility of active transport mechanisms, which needs further investigation.

## 5. Plant EV–Microorganism Interactions

In their natural environment, plants and microbes maintain a constant association with each other. The extensive array of microorganisms interacting with plants can have various effects: they may have no discernible impact on plant fitness (neutral), pose a threat as pathogens, or confer benefits as beneficial organisms. This intricate interplay relies on a sophisticated exchange of signals between all parties involved. Both the host plant and microbes release signals as evidence of their presence in this complex ecological dialogue. It is becoming increasingly clear that EVs play a significant role in facilitating communication between plants and their environment. In the study of plant–pathogen interactions, an emerging concept is related to the critical importance of EVs in establishing this inter-kingdom communication.

In a groundbreaking study [5], it was shown that the proliferation of MVBs and their content release in the form of paramural vesicles occurs during the rapid deposition of cell wall appositions in response to *Blumeria graminis* f. sp. *hordei* attacks, or when plasmodesmata between hypersensitive cells and their neighbouring intact cells become blocked. This research clearly illustrates that MVBs serve as complex subcellular structures, possibly containing pre-assembled components for papilla formation and antimicrobial compounds ready for release in defence against pathogen penetration [82].

Recent findings confirmed that bacterial or fungal infection boosts the secretion of plant EVs containing diverse defence-related proteins, sRNAs, and lipid signals, suggesting their involvement in plant defence mechanisms [83,84]. For instance, a study involving Arabidopsis being infected with the powdery mildew *Golovinomyces orontii* demonstrated that plant-derived exosome-like vesicles transport both PEN1/SYP121 and the ABC transporter Penetration 3 (PEN3), which accumulate in the haustorial encasements, creating a defence barrier that restricts fungal entry. Furthermore, in response to infection by the fungal pathogen *Botrytis cinerea*, Arabidopsis cells secreted exosome-like EVs containing sRNAs. Intriguingly, fungal cells internalise these vesicles at the infection sites and uptake specific miRNAs, which leads to the silencing of fungal genes crucial for pathogenicity [64].

From an applied perspective, it is interesting to emphasise that the purified EVs retain their bioactivity.

AVs isolated from the extracellular fluids of sunflower (*Helianthus annuus*) seedlings revealed a heterogeneous population of vesicles containing cell-wall-remodelling enzymes and defence proteins, identified via proteomic analysis. When exposed to purified AVs, the spores of the phytopathogenic fungus *Sclerotinia sclerotiorum* displayed growth inhibition, morphological changes, and cell death [77]. Similarly, De Palma et al. demonstrated that EVs purified from tomato roots effectively suppressed both spore germination and germination tube development in the plant pathogens *Fusarium oxysporum*, *B. cinerea*, and *Alternaria alternata* [21].

While the EV-mediated interactions between plants and fungi have been studied, bacteria–plant interactions remain relatively unexplored. Rutter and Innes [37] observed that *Pseudomonas syringae*-infected Arabidopsis plants expressed doubled numbers of EVs compared to uninfected controls. This finding strongly implies that plant EVs might play a crucial role in conferring resistance against bacterial pathogens. Moreover, the EVs isolated from *Arabidopsis* infected with *P. syringae* were found to contain numerous signal transduction proteins associated with biotic stress, suggesting that plant EVs could also serve as carriers of immune signals during bacterial infections.

Collectively, these data demonstrate how fungal EVs are employed to evade plant defence and highlight the crucial role of EVs in the bidirectional communication between plants and fungi in pathogenic contexts. However, one of the most intriguing aspects, perhaps yet to be unravelled, is the role of EVs in the mutualistic and symbiotic relationships between organisms. Even if the role of the plant endomembrane system in EV biogenesis, trafficking, and uptake has been recently reviewed, there are still missing data about the real role of EVs during arbuscular mycorrhizal symbiosis [84]. A recent ultrastructural analysis documented the presence of EVs within the peri-arbuscular interface during the formation and maturation of *Rhizophagus irregularis* arbuscules in rice (*Oryza sativa*) [85]. Recently, common bean (*Phaseolus vulgaris*) *PvTET8.1*, the homologue of the *A. thaliana* exosomal marker TET8, was found to be upregulated during arbuscular mycorrhizal fungi and rhizobia association [84]. Moreover, the silencing of *pvTET8.1* reduces the size and the number of nodules, nitrogen fixation, and mycorrhizal arbuscules. However, the physical and functional involvement of EVs in symbiotic plant–microbe interactions needs to be demonstrated. Therefore, research on these aspects is of utmost interest, also considering the already well-established data obtained in very different contexts, such as the gut microbiota, where EVs appear to be essential for shaping the microbial community, thus contributing to the maintenance of a healthy intestinal microbiota.

## 6. Plant EV–Virus Interactions

RNA viruses utilise the host’s cellular membranes to support every step of their life cycle, including the intracellular compartmentalisation of replicases, movement to neighbouring cells, and sheltering from the host’s antiviral immune system [86]. Rearrangements may lead to the formation of spherules, vesicles, or MVBs, often bound by a double-layer membrane and connected to the cytosol using small channels [87]. These different sorts of vesicles, sometimes described as replication factories, are supposedly generated to create a protective microenvironment for virus replication and for the consequent production of new virions. Virus families, genera, or even individual species within a genus engage different endomembrane types in a selective way to exert their functions. The biological significance of this wide organellar diversity that viruses interact with is still the matter of study and speculations.

Certain plant positive-sense single-stranded RNA may modify existing membranes, generating typical spherules, usually in the shape of invaginations of the external membranes of peroxisomes, mitochondria, or chloroplasts [88,89,90]. However, the most targeted endomembrane by this group of viruses is the endoplasmic reticulum (ER), which has been shown to harbour vesicular structures ranging from 30 to 300 nm in diameter upon virus infection. Viruses eliciting ER-derived vesicles belong to different families, including, to name the most studied, Alphaflexiviridae (e.g., potato virus X, PVX), Potyviridae (turnip mosaic virus, TuMV), Secoviridae (grapevine fan leaf virus, GFLV), and Virgaviridae (tobacco mosaic virus, TMV) [87,88,89,90]. Whereas virus-generated membrane spherules are static, ER-derived vesicle-shaped viral factories are motile, and have an important role in the intracellular movement of viral nucleic acids and proteins. The intracellular mobility of vesicles containing viral RNA and proteins exploits the actin network and cytoskeletal organisation [91,92].

Recent findings have suggested the idea that vesicles are key factors in plant–virus interactions and can represent more than simple shelters for virus replication. A study by Movahed and colleagues [18] was a breakthrough for understanding the involvement of vesicle-mediated intracellular and extracellular transport in the course of viral infection. The authors demonstrated using confocal microscopy and TEM that TuMV proteins, e.g., the membrane-bound 6K2, travelled within EVs in the apoplast space outside the plasma membrane in infected *N. benthamiana* leaves. Proteomic analysis of the EV extracts confirmed the presence of TuMV proteins in the extracellular space of the plant leaves [18]. This observation seems in contrast with the traditional view of viruses as obligate intracellular molecular parasites that are confined within living cells throughout their entire life cycle. Mammadova et al. [13] analysed tomato NVs, i.e., a heterogeneous mixture of intra- and extracellular vesicles, as well as the vesicles that formed in the isolation process, extracted via tissue homogenisation from tomato leaves. Combined application of MS-based proteomics and cryoTEM analysis identified six different tomato viruses and their proteins in the low-density fractions of gradient-density ultracentrifugation-isolated vesicle populations, which showed high morphological similarity with mammalian EVs. Based on those observations, the authors hypothesised that the identified viral proteins could be secreted extracellularly within the NVs. Another study showed that viral particles were present in the apoplast fluid isolated from *N. benthamiana* plants infected with PVX, and the apoplast with viral particles was infectious via rub inoculation for healthy plants. The PVX coat protein (CP) was the prevalent viral protein, while viral nucleic acid was detected using reverse transcription-PCR and Northern blot. However, by isolating the exosomes (AVs) from the apoplastic fluid, PVX was detected outside but not inside the vesicles, suggesting that the intact PVX virions do not share the vesicle-mediated bidirectional transport to and from the extracellular space [93].

## 7. Conclusions

Plant EVs have emerged as crucial mediators in cell–cell communication processes and inter-kingdom interactions. However, it is important to acknowledge that research in this field is still at its early stages. Several critical aspects need to be addressed to deepen our understanding of plant EVs and harness their potential for innovative applications in agriculture, biotechnology, and crop protection. This includes our progress in understanding the biogenesis mechanisms of plant EVs. Utilising genetic mutants in the exploration of candidate pathways for EV formation and secretion can yield invaluable insights into the complex mechanisms governing EV generation and release. Furthermore, this research will be instrumental in elucidating whether the loading of specific biomolecules into plant EVs is driven by an active and selective mechanism, a passive process, or a combination of both, thereby enhancing our comprehension of their regulation and potentially control their release. Upcoming research might show how the cargo of plant EVs changes in response to different host–microbe interactions, including those with beneficial symbiotic organisms and pathogens. Understanding the dynamic nature of cargo loading in EVs will also help elucidate their roles in plant stress responses, defence mechanisms, and other physiological processes.

Another intriguing and yet poorly understood aspect is the long-distance gene expression regulation mediated by plant EVs. It is widely accepted that mobile siRNAs and miRNAs function as established signalling molecules over long distances in multicellular organisms, including plants. However, our understanding of the roles played by mobile mRNAs within recipient cells and tissues, the mechanisms governing its selection for transport and uptake, and the potential involvement of EVs in facilitating these processes remains relatively unexplored.

Moreover, new studies on EV sRNA shuttling will allow the development of innovative delivery methods of sRNA for novel disease control strategies against pathogens and pests in agriculture, and for therapeutic applications in mammalian systems. 

As final remark, we believe that the study of plant EVs presents a vast and promising field of research with a multitude of untapped opportunities. Multidisciplinary collaborative initiatives are crucial to fully explore their potential and address unanswered questions in this field.

## Figures and Tables

**Figure 1 plants-12-04141-f001:**
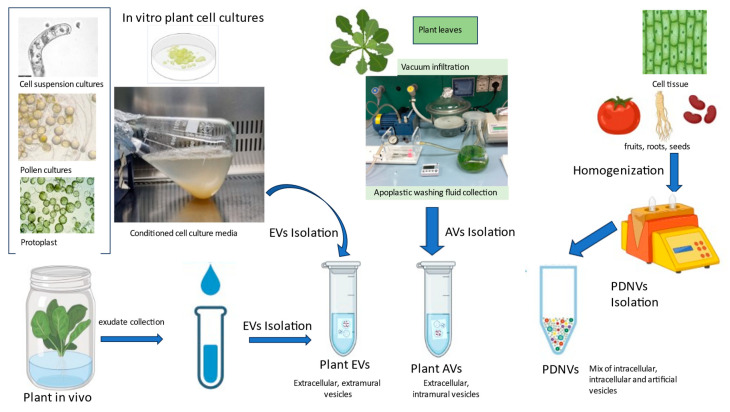
Sources of plant extracellular vesicles (EVs), apoplastic vesicles (AVs), and plant-derived nanovesicles (PDNVs). Plant EVs are extramural (outside the cell wall) and can be isolated from conditioned media of in vitro cell cultures or exudates (like root exudate) that plants release into the environment. AVs are a heterogeneous class of vesicles including intramural vesicles that pass through the cell membrane and nanovesicles found in the phloem and xylem. AVs are generally isolated from apoplastic washing fluids after vacuum infiltration of plant tissues. PDNVs are isolated from homogenate of plant tissue or organs (fruit, leaves, roots, and seeds). A PDNV isolate inherently contains a (i) complex set of membrane-bound intracellular vesicles (transport vesicles, secretory vesicles) that comes from the rupture of the plant cells, as well as (ii) EVs, (iii) AVs, and (iv) vesicles that are formed due to the homogenisation process. These valuable biomaterials are studied for their positive effect on human health or as delivery vectors. Plant EVs and AVs, the subject of this review, are isolated from the conditioned culture media (CCM) of plant in vitro cultures, apoplastic fluid, or root exudate.

**Figure 2 plants-12-04141-f002:**
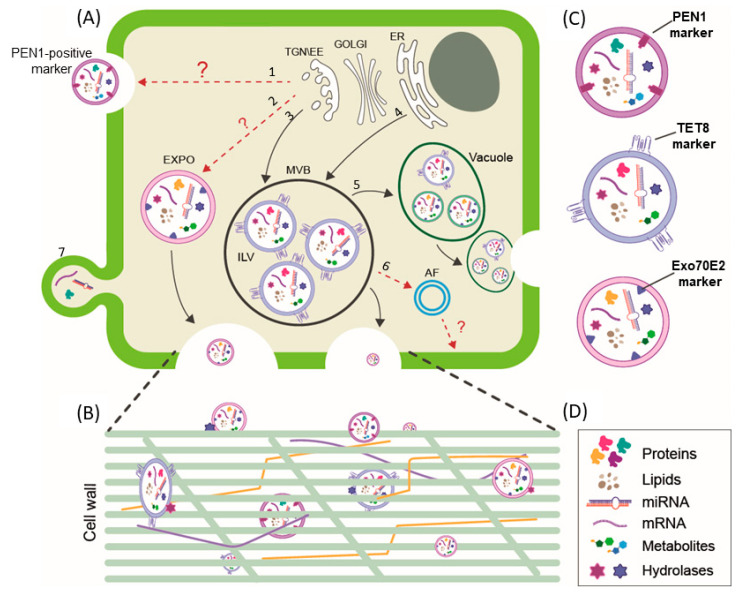
Plant EV biogenesis and release. (**A**) Possible EV biogenesis pathways in plants. Plant EVs might be derived from PEN1-positive organelles (1); exocyst-positive organelles (EXPOs) (2); multivesicular body (MVB) endosomes (3); ER-derived vesicles (4); vacuole fusion with the plasma membrane (5); autophagosome-mediated secretion (6); or blebbing of the cell membrane (7). (**B**) Passage of EVs through the plant cell wall might involve cell wall hydrolases. Cell wall as well as EV plasticity could further facilitate their cell wall crossing. (**C**) Schematic representation of biomarker-associated EVs. (**D**) Plant EV composition. TGN/EE: trans-Golgi network/early endosome; ER: endoplasmic reticulum; EXPO: exocyst-positive organelle; MVB: multivesicular body; ILV: intraluminal vesicle; AF: autophagosome. ? means putative route.

**Table 1 plants-12-04141-t001:** Protein, small RNA (sRNA), lipid, and metabolite cargo characterisation in different plant-derived EVs.

Plant Species	Source/Tissue	Biocargo/Putative Function	References
Protein analysis			
*Olea europaea*	Germination medium/pollen grains	Proteins associated with metabolism and signalling, cell wall expansion, defence, and stress response	[38]
*Actinidia chinensis* Planch.	Germination medium/ pollen grains	Proteins involved in metabolic processes, transport, signalling, and stress response	[41]
*Arabidopsis thaliana*	Apoplastic washing fluid/leaves	Proteins involved in metabolic processes, cell wall organization, and biotic and abiotic stress responses	[25,37]
*Arabidopsis thaliana*	Apoplastic washing fluid/leaves	Proteins involved in stress response, RNA binding proteins involved in sRNA selective loading and stabilisation in EVs	[67]
*Helianthus annus*	Extracellular fluid/seedlings	Proteins involved in cell wall modification, defence, vesicular trafficking events	[77]
*Nicotiana benthamiana*	Apoplastic washing fluid/leaves	Proteins involved in cell communication, metabolic processes, transport, and stress response	[18]
*Solanum lycopersicum*	Collection medium/root	Proteins involved in perception and transduction of plant–pathogen interactions and defence-related proteins	[21]
*Salvia dominica*	Culture media/hairy root	Cytoskeletal components, cell wall organisation, chaperon proteins	[42]
*Helianthus annus*	Extracellular fluid/seeds	Small GTPase as regulators of vesicular trafficking events	[34]
*Craterostigma plantagineum* Hochst.	Cell suspension culture media	Proteins involved in cell wall remodelling, defence response	[23]
RNA analysis			
*Arabidopsis thaliana*	Apoplastic washing fluid/leaves	Diverse species of sRNA (tiny RNA, miRNA, and siRNA	[78]
*Arabidopsis thaliana*	Apoplastic washing fluid/leaves	Diverse sRNA species and (long noncoding RNAs, sRNA, and circular RNA) putatively involved in host-induced gene silencing	[33]
*Arabidopsis thaliana*	Apoplastic washing fluid/ homogenised callus	miRNAs, ESCRT complexes, PEN1 and TET8	[48]
Lipid and Metabolite analysis			
*Helianthus annus*	Extracellular fluids of imbibed sunflower seeds	Detection of phosphatidic acid (PA), phosphatidylinisitol (PI), phosphatidylethanolamine (PE), and phosphatidylcholine (PC)	[34,79]
*Arabidopsis thaliana*	Apoplastic washing fluid/leaves	Presence of sphingolipids, phospholipids, and sterols	[27]
*Nicotiana tabacum*, *C. plantagineum* Hochst.	Apoplastic fluid/suspension culture media	Enrichment of lipophilic compounds	[23]
*Aster yomena*	Cell suspension culture medium	Seventeen metabolites	[46]

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
