# Peer review of "Plant Extracellular Vesicles: Current Landscape and Future Directions"

_plants, 2023, doi:10.3390/plants12244141_

Round 1

Reviewer 1 Report

Comments and Suggestions for Authors

The review is devoted to plant extracellular vesicles (EVs) carrying functional molecules, which, like their animal counterparts, are suggested to participate in numerous physiological and pathological processes. In particular, the authors have focused the final part of the review on the possible roles of EVs as mediator in plant pathogen interactions. Recent reviews have been yet published on this topic.

Here are listed some detailed areas of concern.

This present review is based on 84 references but I regret that only 2 references are published in 2023 and a thirty in 2021-2022.

For example, the following references published in 2023 are missing from the review: Jang et al. 2023. Front Plant Sci. 13:1064412. doi: 10.3389/fpls.2022.1064412 ; Yugay et al (2023). Plants 12(20):3604. doi: 10.3390/plants12203604 ; Parra-Aguilar et al (2023). Front Plant Sci. 3;14:1152493. doi: 10.3389/fpls.2023.1152493 ; Holland & Roth (2023). Mol Plant Microbe Interact. 36(4):235-244. doi: 10.1094/MPMI-09-22-0189-FI ; Abubakar et al (2023). Stress Biol. 2023 Aug 22;3(1):35. doi: 10.1007/s44154-023-00114-0 ; Cheng et al (2023). Appl Microbiol Biotechnol. 107(19):5935-5945. doi: 10.1007/s00253-023-12718-7

The short introduction promptly describes the history of plant EVs, their ways of secretion into the extracellular space and their sources of production. This paragraph is associated with a simplistic figure (fig 1) summarizing the sources and types of vesicles: extracellular vesicles (EVs), apoplastic vesicles (AVs), or nanovesicles/plant derived-nanovesicles (NVs/PDNVs). The latter type is not clear as it does not appear to be “extracellular”. One may ask whether this figure is really informative. Photos are tiny and not described, the term “nanovesicles vesicles” is used, tubes containing EV and AVs look identical. An effort should be done to reconstruct this figure. In addition, the legend is redundant with respect to the introduction and both therefore should be remodelled.

The review continues in an interesting way with a first paragraph dedicated to in vitro cultures as sources of EVs such as cell suspension cultures, protoplasts, hairy roots (only a single article) and pollen cultures with many experimental details and characteristics (size, protein or lipid markers...) of EVs isolated from these various sources.

The following paragraph presents EV biogenesis and release, emphasizing the central role of multivesicular bodies (MVBs), and briefly introducing alternative EV biogeneses including exocysts, autophagosomes, and vacuole-PM fusion. The figure 2 is well constructed. Some question may still be raised: what is the origin of EXPO? Is endocytosis at the origin of EV (empty circle)? How PEN-decorated vesicles are released (recycling?)? How are vesicles not degraded in lytic vacuoles?  As shown in line 511, there are also ER-release vesicles that can be added.

A “3.1” paragraph exists (and sub-paragraphs) but there is no “3.2”. I propose to change logically the numbering of “Plant EV Biocargo” paragraph to “4” instead of “3.1” and so on.

This paragraph presenting EV biocargoes and their function is well described and accompanied by a comprehensive table.

Finally, the last two parts are focused on plant-microbe interactions (including mutualist interactions) and plant-virus interaction, opening up a host of questions as many correlations are found, underlighting that the knowledge of the underlying mechanisms are still in their infancy and relatively unexplored.

Despite the criticisms, which the authors can consider to improve it, the review has the advantage of covering different aspects related to these naturally plant secreted vesicles, particularly during biotic interactions whose mechanisms are not yet well understood and relatively not well explored both in agroecology to pest control or in therapeutic application for human health.

Minor points:

A list of abbreviations will be welcome.

Homogenate the use of NVs and PDNVs, and explain better their interest.

Some paragraph could be reshaped to clarify some parts or make them more relevant. For example, lines 99 to 104 could be transferred to plant-microbe interaction paragraph.

Reviewer 2 Report

Comments and Suggestions for Authors

The manuscript reviewed the in vitro sources that have been used to isolate plant EVs and the studies that investigate the molecular cargo and pinpoints possible roles of plant EVs as mediator in plant pathogen interactions. The paper provided latest and comprehensive information of plant EVs, which is useful for the subsequent studies of researchers on plant EVs.

Some minor comments:

There have been several reviews of plant EVs published in recent years. What are the differences of your reviews?

Reviewer 3 Report

Comments and Suggestions for Authors

This is a well written manuscript and clearly embodies considerable effort.  However, as a 'review' (how you authors elected to classify it), it could and should make a more substantive contribution.

A major and, in my opinion, the primary purpose of a 'review' is to help readers, especially graduate students and others entering the field of research, become aware of the true state of knowledge, including solid vs questionable methodologies and other issues needing of resolution through future research.

What readers hope to find in a useful review are 1) clear definition of terms, 2) comprehensive historical treatment about the development of the field with appropriate crediting of everyone, 3) rigorous critique of the current state of knowledge with emphasis on unresolved areas, possible artifacts and other uncertainties, 4) full details about methodologies and the underlying assumptions attending them, 5) identification of contentious issues, and 6) suggestions for future research.   Although touching on those, this review fails to make a solid contribution in any of those six areas.

There is little about the original discoveries leading to the concept of MVBs, only one quite unacceptably brief mention of yeast in vitro research, no mention of the pioneering work with algae, no mention of clathrin coating, no TEM of biochemical images to help readers understand the variable nature of EVs, no mention of high-pressure freezing.  The 'paramural bodies' mentioned in the Introduction have long been regarded as fixation artifacts, but this ongoing concern is not addressed. 

Figure 2 illustrates some important concepts; however, those concepts are not adequately fleshed out within the review.  It seems that the authors have written for those in the know rather than for those who are entering the field - raising the question of why bother publishing it at all. The defining feature of MVBs is the presence of intra-lumenal vesicles, but this review does little to explain MVB (variable) origins or clearly how vesicles arise within MVBs. 

This paper is attempting to address a highly important topic.  Considering all of the above comments (and many more could have been made), I suggest you revise this paper into a truly useful contribution.

Comments on the Quality of English Language

well written

Round 2

Reviewer 1 Report

Comments and Suggestions for Authors

All comments have been taken into account by the authors. They have modified or completed the figures. They have added recent publications, explaining their choice from among those I have suggested. This improves the clarity and focus of the review. Most other concerns have also been addressed.

I invite the authors to reread the text very carefully.

Reviewer 3 Report

Comments and Suggestions for Authors

This is a useful narrative of what we think we know.  It could be more critically revealing of the assumptions underlying the topic.